

# Outcomes of stereotactic body radiotherapy for unresectable cholangiocarcinoma: a meta-analysis and systematic review

Peng Liu[1,*], Haiyan Ye[2,*], Lijun Song[2], Hua Li[2], Mingna Fu[2] and Zhichao Dong[1]

[1] Department of Radiotherapy, Yantai Yuhuangding Hospital, Yantai, China
[2] Department of Oncology, Laizhou People's Hospital, Yantai, China
[*] These authors contributed equally to this work.

Corresponding authors
Mingna Fu, f935@163.com
Zhichao Dong, dzcytyhd@163.com

## ABSTRACT

**Background.** Stereotactic body radiotherapy (SBRT) is an effective treatment for various malignancies. This meta-analysis aimed to determine the prognostic outcomes and toxicities of SBRT for unresectable cholangiocarcinoma (CC) using the most recent evidence.

**Methods.** The review protocol was registered on PROSPERO (CRD42023393642). We searched PubMed, EMBASE, and the Cochrane Library for studies involving SBRT and CC. Study endpoints included median overall survival (OS), 1- and 2-year OS rates, 1- and 2-year disease control rates (DCR), and the incidence of grade 3 or higher toxicities.

**Results.** Thirteen studies (366 patients) were included. Of these, 138 patients (37.7%) had extrahepatic CC and 228 patients (62.3%) had intrahepatic CC. The median total SBRT dose was 45 Gy, and the median biologically effective dose ($BED_{10}$) was 72.0 Gy. The pooled median OS was 13.4 months (95% confidence interval (CI) [10.9–15.8]). Pooled 1-year and 2-year OS rates were 58.7% (95% CI [53.8–63.7%]) and 33.2% (95% CI [28.3–38.2%]), respectively. Pooled 1-year and 2-year DCR rates were 84.7% (95% CI [81.0–88.3%]) and 70.5% (95% CI [65.2–75.8%]), respectively. Pooled incidence rates for grade ≥3 acute, late, and overall toxicity were 6.4% (95% CI [2.6–10.1%]), 16.4% (95% CI [1.9–31.0%]), and 16.9% (95% CI [9.3–24.6%]), respectively. No factor was significantly associated with improved OS or DCR.

**Conclusions.** This meta-analysis demonstrates that SBRT may be an efficacious and safe therapeutic option for unresectable CC. Further prospective studies comparing SBRT with alternative treatment approaches are required to define its definitive role in managing CC.

# INTRODUCTION

Cholangiocarcinoma (CC) accounts for 3% of gastrointestinal (GI) malignancies and represents the second most prevalent primary liver malignancy (*Sung et al., 2021*). Surgical resection with negative histological margins remains the only potentially curative

treatment. However, most patients present with advanced disease at diagnosis and are therefore ineligible for surgery; their reported median prognosis is less than seven months (*Shinohara et al., 2008*; *Valle et al., 2021*). For these patients, non-surgical therapies—including radiation therapy (RT), photodynamic therapy, and chemotherapy—have been utilized, though outcomes remain unsatisfactory (*Kahaleh et al., 2008*; *Patel, 2011*), and no preferred standard approach has been established.

RT is an effective local treatment modality for many cancers, but conventional RT delivers suboptimal results for CC due to its inability to administer ablative tumor-control doses (*Chen et al., 2010*; *Zeng et al., 2006*). This limitation stems from the proximity of radiosensitive bowel structures to CC lesions and their low tolerance to radiation. Crucially, a higher biologically effective dose (BED) correlates significantly with improved overall survival (OS) (*Deodato et al., 2006*; *Morganti et al., 2000*). Thus, advancing precision radiotherapy techniques is essential to escalate tumor doses while sparing adjacent organs.

Technical innovations in delivery precision and respiratory motion management over the past decade have enabled the widespread adoption of stereotactic body radiation therapy (SBRT), which delivers ablative radiation doses in one or few fractions. Recent prospective studies of SBRT for hepatocellular carcinoma demonstrate promising efficacy, with 1-year local control rates of 86–95% (*Bujold et al., 2013*; *Kang et al., 2012*; *Lasley et al., 2015*). Outcomes of SBRT for CC have also been reported, yet its role remains controversial. Systematic reviews by *Frakulli et al. (2019)* concluded that robust evidence supporting SBRT in CC is lacking, whereas *Lee et al. (2019)* found it to be a feasible option, achieving high local control in unresectable or recurrent disease. Recent publications have further contributed evidence (*Brunner et al., 2019*; *Kozak et al., 2020*; *Liu et al., 2017*; *Thuehøj et al., 2022*; *Zhang et al., 2022*). To clarify the efficacy and safety of SBRT for unresectable CC, we conducted this systematic review and meta-analysis of published studies, focusing on survival outcomes and toxicity.

## MATERIALS AND METHODS

### Search strategy

The Population, Intervention, Control, Outcomes, and Study Design (PICOS) approach was adopted to set the inclusion criteria. We systematically searched the Cochrane Library, Embase, PubMed, and MEDLINE databases until February 19, 2024. The following patterns were employed to find articles on CC and SBRT: (cholangio or bile or Klatskin or hilar) AND (SBRT or SABR or stereotactic body radiation or stereotactic ablative body radiotherapy). Only the articles in English were searched and checked. This review protocol is registered on PROSPERO (the International Prospective Register of Systematic Review, CRD42023393642).

### Study selection

The inclusion criteria were: (1) published in English; (2) reported locally advanced CC; (3) prospective or retrospective studies or clinical trials including at least ten patients; (4) reported OS, DCR, and toxicity data. After inclusion criteria were applied to the search result, the studies that met the following criteria were excluded: (1) studies involving

metastatic or re-irradiation cases; (2) studies with cohort selected from the online database; (3) mixed with other tumors and the clinical data that could not be distinguished; (4) reviews, meta-analysis, *in vitro* or animal studies, editorials, and protocols. Lastly, a full-text review was conducted to verify that the articles met the inclusion and exclusion criteria. Two separate researchers carried out the aforementioned procedures, and the outcomes of their searches were consistent.

## Data extraction

Data was taken from the included publications by two scholars (PL and HY). The other co-authors settled any disputes that could have arisen. The following data were extracted: author name, publication year, country, age of patients, total cases, the number of patients with intrahepatic cholangiocarcinoma, tumor size, chemotherapy history, total SBRT dose, biological effect dose (BED) 10, number of fractions, DCR (1- and 2- years), median overall survival (OS), OS (1- and 2- years), and the incidence of grade 3–5 toxicities (acute GI, hematologic and late GI, hematologic toxicities). If the OS was not described in detail (for example, the range of OS), the data were extracted from the Kaplan–Meier (K-M) curves using Engauge Digitizer V11.4. The BED was used to assess the equivalence in biological effects between different fractionated doses and irradiation protocols. If the BED was not reported, the value was calculated as follows: $BED = d * [(1 + (d/n \div \alpha/\beta)]$. The $\alpha/\beta$ ratio reflects differences in tissue sensitivity to radiation damage. Most tumor cells are proliferatively active (similar to early-responding tissues like skin and mucosa), exhibit weaker repair capacity, and are thus more sensitive to variations in fractionated dose. Experimental and clinical data have shown that the $\alpha/\beta$ value of tumors typically approximates 10 Gy. Thus, in our study, the $\alpha/\beta$ ratio was set at 10 ($BED_{10}$) (n = number of fractions, d = total dose) (*Fowler, 1989*). Any disagreements among co-authors for data extraction were resolved by the corresponding author Z.C.D.

## Quality assessment

Due to most of the pertinent research being retrospective or single-arm studies, we employed the Newcastle Ottawa Scale (NOS) as an evaluation indicator. Studies were categorized as medium quality if their NOS rating was 4–6 and as high quality if their NOS rating was 7–9.

## Statistical analysis

The primary endpoints in this meta-analysis were the DCR, OS, and treatment-related toxicity. The relationship between $BED_{10}$ for DCR and toxicity was also analyzed. The above endpoint proportions were pooled and analyzed. The $I^2$ value was used to assess the heterogeneity between the studies. A random effect result was used with an otherwise fixed-effect outcome when $I^2 > 75\%$ was judged to be quite heterogeneous. A sensitivity analysis was conducted to identify potential studies that could contribute to notable heterogeneity. Publication biases were assessed quantitatively using Egger's test of the intercept and visually using the funnel plot. Meta-regression models were utilized to assess the relationship between $BED_{10}$ and each respective outcome (DCR, OS, and toxicity).

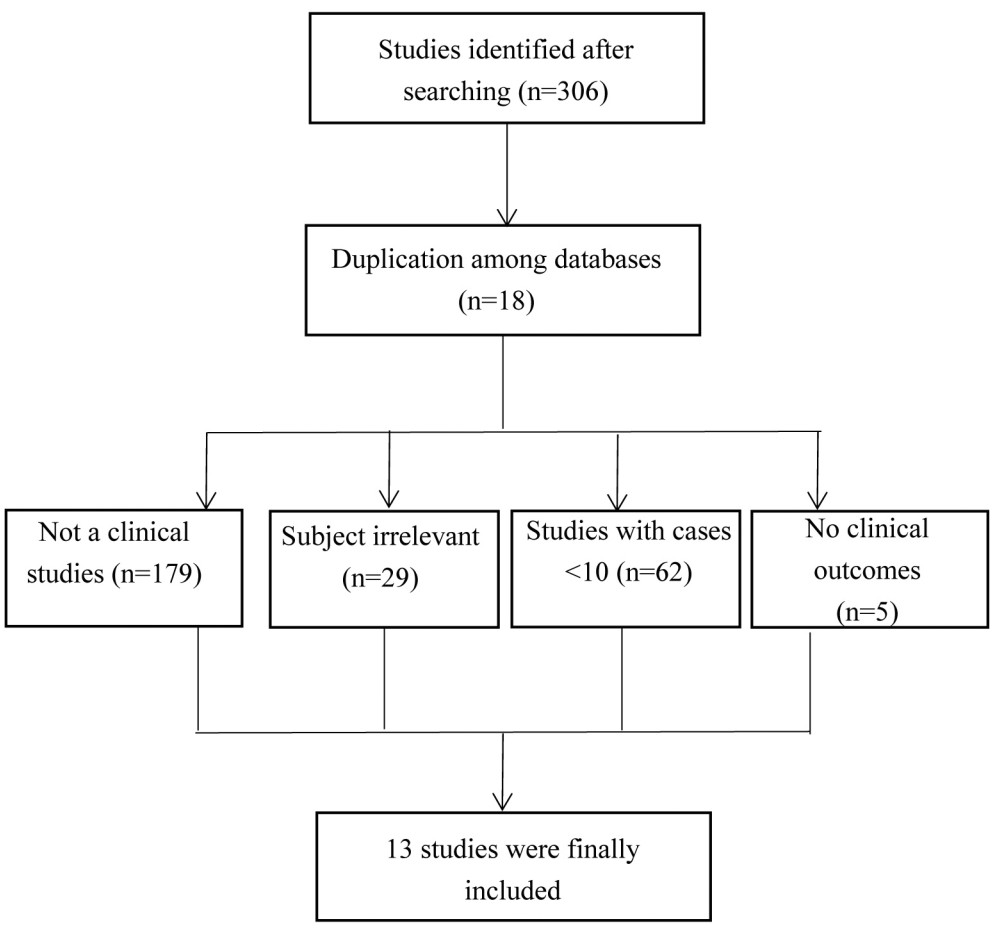

**Figure 1 The flow chart of study selection.**

Stata (Version 15.0) was used for all statistical analyses. Statistical significance was defined as a *P* value less than 0.05.

# RESULTS

## Search results and publication bias

A total of 308 studies were extracted. After study screening, 13 studies were finally included in our study (*Brunner et al., 2019*; *Ibarra et al., 2012*; *Kopek et al., 2010*; *Kozak et al., 2020*; *Liu et al., 2017*; *Mahadevan et al., 2015*; *Polistina et al., 2011*; *Sandler et al., 2016*; *Shen et al., 2017*; *Thuehøj et al., 2022*; *Tse et al., 2008*; *Welling et al., 2014*; *Zhang et al., 2022*). The process for study inclusion is presented in Fig. 1. The included studies are presented in Table 1.

The funnel plot and Egger's test did not show publication bias in the pooled analysis of OS ($P = 0.795$, Fig. S1).

Liu et al. (2025), *PeerJ*, DOI 10.7717/peerj.19909

**Table 1 Characteristics of the included studies.** Note. *Zhang et al., 2022*; *Liu et al., 2017*; *Thuehøj et al., 2022*; *Tse et al., 2008*; *Mahadevan et al., 2015*; *Brunner et al., 2019*; *Kozak et al., 2020*; *Ibarra et al., 2012*; *Sandler et al., 2016*; *Kopek et al., 2010*; *Polistina et al., 2011*; *Shen et al., 2017*; *Welling et al., 2014*.

| Author | Publication year | Study design | Country | Median age (range), y | Total patients (n) | Tumor location Intrahepatic/ extrahepatic, n | Tumor size (mm), median, range | With chemotherapy history, (n) | Timing of chemotherapy | Total SBRT dose, median, Gy | BED$_{10}$, Median | Fraction, median (range) |
|---|---|---|---|---|---|---|---|---|---|---|---|---|
| Zhang et al. | 2022 | R | China | <60 (n = 16) >60 (n = 27) | 43 | 43/0 | | | | 40 | 63 | 7 |
| Liu et al. | 2017 | R | China | 74 (53–86) | 15 | 13/2 | 42 (12–130) | 15 | Concurrent | 45 | 72 | 5 |
| Thuehoj et al. | 2022 | R | Denmark | 69 (39–82) | 41 | 15/26 | 22 (5–65) | 5 | Before | 48 | 96 | 4.8 (3–6) |
| Tse et al. | 2008 | P | Canada | 57 (49–79) | 10 | 10/0 | | 4 | Before | 32.5 | 57.6 | 6 |
| Mahadevan et al. | 2015 | R | USA | 72 (38–94) | 34 | 34/0 | | 18 | Concurrent | 30 | 60 | 3 |
| Brunner et al. | 2019 | R | Germany | 64 (36–90) | 64 | 41/23 | | 27 | After | 42 | 67.2 | 8 (3–17) |
| Kozak et al. | 2020 | R | USA | 71 (45–89) | 40 | 26/14 | 42 (10–125) | 10 and 17 | Before and after | 40 | 72 | 5 (1-5) |
| Ibarra et al. | 2012 | R | USA | 66 (43–86) | 11 | 11/0 | | 5 | before | 30 | 60 | 3 (1–10) |
| Sandler et al. | 2016 | R | USA | 63 (32–94) | 31 | 6/25 | 27 (10–73) | 23 | before | 40 | 72 | 5 |
| Kopek et al. | 2010 | R | Denmark | 69 (35–86) | 27 | 1/26 | | | | 45 | 112.5 | 3 |
| Polistina et al. | 2011 | R | Italy | 69.5 (49–75) | 10 | 0/10 | | 10 and 10 | Before and after | 30 | 60 | 3 |
| Shen et al. | 2017 | R | China | <69 (n = 11) >60 (n = 17) | 28 | 28/0 | <50 (n = 6) 50–100 (n = 15) >100 (n = 7) | 8 | before | 45 | 112.5 | 5 (3–5) |
| Welling et al. | 2014 | R | USA | 58 | 12 | 0/12 | 14.2 (0–32) | 12 | | 55 | 140 | 4 (3–5) |
| Pooled | | | | | 366 | 228/138 | 30 (0–130) | | Before: 65 Concurrent: 33 After: 54 | 40, range from 30–55 | 72, range from 57.6–140 | 5, range from 3–8 |

**Notes.**

P: prospective study; R: retrospective studies; SBRT: stereotactic body radiation treatment; BED: biological effect dose.

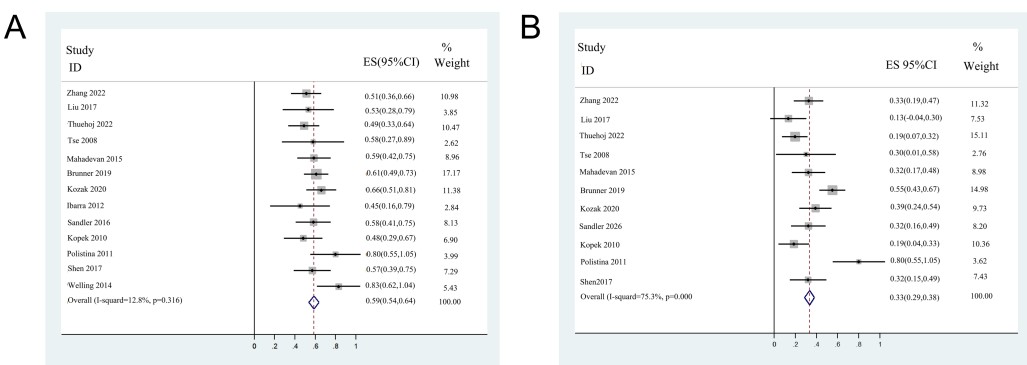

**Figure 2 Forest plots of overall survival rates.** (A) 1-year overall survival rates (fix-effect model). (B) 2-year overall survival rates (random-effect model).

## Study characteristics

Only one study (7.7%) had a prospective design (*Tse et al., 2008*), and the other 12 studies (92.3%) were retrospective. According to the NOS, all 13 studies were of medium quality (12 studies scored 6 points and 1 study scored 5 points). The quality evaluations of all included articles are in Table S1.

A total of 366 patients with locally advanced CC were included in our study. Among them, 138 patients (37.7%) were identified as having extrahepatic CC, and 228 patients (62.3%) as having intrahepatic CC (ICC). Only five studies reported detailed tumor size. The pooled tumor size was 30 mm, ranging from 0 to 130 mm. The chemotherapy protocol was described in all studies except two (*Kopek et al., 2010*; *Zhang et al., 2022*). Not all studies reported the chemotherapeutic regimens; however, the most frequently described regimens were gemcitabine and 5-FU. Before SBRT, 65 patients (65/296, 22.0%) underwent chemotherapy, and 33 (33/296, 11.1%) underwent concurrent chemotherapy. After SBRT, 54 patients (54/296, 18.2%) underwent chemotherapy. The total SBRT dosage ranged from 30 to 55 Gy, with a median of 45 Gy. The $BED_{10}$ was calculated for each study. The estimated $BED_{10}$ range is 57.6–140, with a median value of 72.0. The number of fractions ranged from three to eight among studies, with a median of 5. The characteristics of the research are enumerated in Table 1.

## Pooled analysis of OS

Among the 13 included studies, 12 reported detailed data on OS, except for one (*Welling et al., 2014*). For the remaining 354 CC patients who underwent SBRT, the pooled median OS was 13.4 months (95% confidence interval (CI) [10.9–15.8]). The 1-year OS was reported in all studies and 2 years for 11 studies. Moreover, three 1-year and two 2-year OS were extracted from the K-M curves. The median 1-year OS ranged from 45–83%, with a pooled 1-year OS of 58.7% (95% CI [53.8–63.7%]). The median 2-year OS ranged from 13.3–80%, with a pooled 2-year OS of 33.2% (95% CI [28.3–38.2%]). The forest plots of 1- and 2-years OS are shown in Fig. 2.

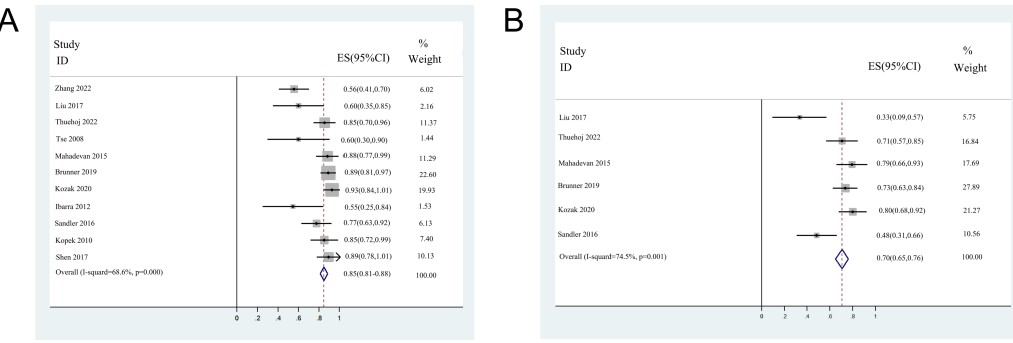

**Figure 3  Forest plots of disease control rates.** (A) 1-year disease control rates (random-effect model). (B) 2-year disease control rates (random-effect model). Note. *Zhang et al., 2022*; *Liu et al., 2017*; *Thuehøj et al., 2022*; *Tse et al., 2008*; *Mahadevan et al., 2015*; *Brunner et al., 2019*; *Kozak et al., 2020*; *Ibarra et al., 2012*; *Sandler et al., 2016*; *Kopek et al., 2010*; *Polistina et al., 2011*; *Shen et al., 2017*; *Welling et al., 2014*.

## Pooled analysis of DCR

The 1-year DCR was reported in 11 studies, except for two small sample size studies due to the insufficient follow-up time (*Tse et al., 2008*; *Welling et al., 2014*). The 2-year DCR was reported in six studies, including one DCR extracted from the K-M curve. The median 1-year DCR ranged from 54.6 to 92.5%, with a pooled 1-year DCR of 84.7% (95% CI [81.0–88.3%]). The median 2-year DCR ranged from 33.3 to 80%, with a pooled 2-year DCR of 70.5% (95% CI [65.2–75.8%]). The forest plots of 1- and 2-year DCR are shown in Fig. 3.

## Toxicity

Every study reported post-radiation toxicities. However, some studies did not grade the toxicities in detail (*Thuehøj et al., 2022*; *Welling et al., 2014*). The incidence of acute toxicities ranged from 0 to 30%, with pooled acute toxicities of 6.4% (95% CI [2.6–10.1%]). Late toxicities ranged from 0 to 37.5%, with a pooled late toxicity of 16.4% (95% CI [1.9–31%]). The pooled overall incidence of grade 3 or higher toxicities was 16.9% (9.3–24.6%). Due to the lack of reported toxicities in many studies, we did not create forest plots for the incidence of toxicities. The clinical outcomes of the included patient were summarized in Table 2.

## Heterogeneity and sensitivity analysis

Significant heterogeneity was observed in 2-year OS rate ($I^2 = 75.3\%$, $P < 0.001$), 1-year DCR rates ($I^2 = 68.6\%$, $P < 0.001$), 2-years DCR rates ($I^2 = 74.5\%$, $P = 0.001$), the incidences of late ($I^2 = 83\%$, $P = 0.001$), and overall toxicities ($I^2 = 66.5\%$, $P = 0.004$) among the included studies. The 1-year OS rate ($I^2 = 12.8\%$, $P = 0.316$) and incidence of acute toxicity ($I^2 = 35.9\%$, $P = 0.142$) showed no significant heterogeneity among studies.

For sensitive analysis, the combined 2-year OS rates ranged from 32.8% to 33.5% after omitting any single study. The combined DCR rates for 1-year and 2-year ranged from 81.0% to 88.3% and 64.7% to 76.2%, respectively, after omitting any single study. The combined incidence of late and overall toxicity ranged from 10.5% to 21.6% and from

**Table 2 The clinical outcomes of included patient who underwent SBRT.** Note. *Zhang et al., 2022*; *Liu et al., 2017*; *Thuehøj et al., 2022*; *Tse et al., 2008*; *Mahadevan et al., 2015*; *Brunner et al., 2019*; *Kozak et al., 2020*; *Ibarra et al., 2012*; *Sandler et al., 2016*; *Kopek et al., 2010*; *Polistina et al., 2011*; *Shen et al., 2017*; *Welling et al., 2014*.

| Author | N | OS (95% CI), months | 1-year OS, % (95% CI) | 2-years OS % (95% CI) | 1-year DCR % (95% CI) | 2-yeas DCR % (95% CI) | Acute toxicity, n (G ≥ 3), n(%) | Late toxicity (G ≥ 3), n(%) | Total toxicity (G≥ 3), n(%) |
|---|---|---|---|---|---|---|---|---|---|
| Zhang et al. | 43 | 12.0 (3.4–20.6) | 51.2 (36.8–65.4) | 32.6 (20.5–47.5) | 55.6 (41.1–69.6) | NR | 3 (7.0) | 0 (0) | 3 (7.0) |
| Liu et al. | 15 | 12.6 (5.0–19.5) | 53.3 (27.4–77.7) | 13.3 (0.02–41.6) | 60.0 (32.9–82.5) | 33.3 (13.0–61.3) | 0 (0) | 0 (0) | 0 (0) |
| Thuehoj et al. | 41 | 11.8 (3.0–45.0) | 48.8 (34.3–63.5) | 19.5 (10.2–34.0) | 85.4 (71.6–93.1) | 70.7 (57.2–85.2) | NG | NG | NG |
| Tse et al. | 10 | 15.0 (6.5–29.0) | 58 (23.0–82.0) | 30.0 (2.1–58.2) | 60 (27.4–86.3) | NR | 3 (30) | 1 (10) | 4 (40) |
| Mahadevan et al. | 34 | 17.0 (6.2–41.5) | 58.8 (42.2–73.6) | 32.4 (18.0–50.6) | 88.2 (71.6–96.2) | 79.4 (61.6–90.7) | 4 (11.8) | 0 (0) | 4 (11.8) |
| Brunner et al. | 64 | 15.0 (4.0–28.5) | 61.0 (77.0–85.0) | 55.0 (51.0–59.0) | 89.0 (79.1–94.6) | 73.4 (61.5–82.7) | 3 (4.7) | 3 (4.7) | 6 (9.4) |
| Kozak et al. | 40 | 23.0 (15.0–35.0) | 66.0 (52.0–81.0) | 39.0 (23.0–55.0) | 92.5 (78.5–98.0) | 80.0 (65.2–89.5) | 1 (2.5) | 15 (37.5) | 16 (40) |
| Ibarra et al. | 11 | 11.0 (3.0–12.0) | 45 (21.3–72.0) | NR | 54.6 (24.6–81.9) | NR | 1 (9.1) | 0 (0) | 1 (9.1) |
| Sandler et al. | 31 | 15.7 (30.–24.2) | 58.1 (40.8–73.6) | 32.3 (17.3–51.5) | 77.4 (58.5–89.7) | 48.4 (30.6–66.6) | 1 (3.2) | 5 (16.1) | 6 (19.3) |
| Kopek et al. | 27 | 10.6 (4.8–16.3) | NR | 18.5 (7.0–38.8) | 85.2 (65.4–95.2) | NR | 6 (22.2) | 0 (0) | 6 (22.2) |
| Polistina et al. | 10 | 35.5 (12.0–51.0) | 80 (44.2–96.5) | 80 (44.2–96.5) | NR | NR | 0 (0) | 0 (0) | 0 (0) |
| Shen et al. | 28 | 15.0 (7.2–22.8) | 57.1 (37.4–75.0) | 32.1 (16.6–52.4) | 89.3 (70.6–97.2) | NR | 0 (0) | 0 (0) | 0 (0) |
| Welling et al. | 12 | NR | 83 (50.9–97.1) | NR | NR | NR | NG | NG | NG |
| Pooled | 366 | 13.4 (10.9–15.8) | 58.7 (53.8–63.7) | 33.2 (28.3–38.2) | 84.7 (81.0–88.3) | 70.5 (65.2–75.8) | 6.4 (2.6–10.1) | 16.4 (1.9–31) | 16.9 (9.3–24.6) |

**Notes.**

OS: overall survival; DCR: disease control rate; CI: confident interval; NR: not reported; NG: not graded.

10.8% to 22.5% after any single study was omitted. None of the included studies was identified as contributing to heterogeneity.

The included patients were categorized by publication year (before 2016 and after 2016), patient race (Caucasian and Asian), number of patients ($<20$ and $>20$), $BED_{10}$ ($<72$ and $>72.0$), and rates of chemotherapy ($<50\%$ and $>50\%$) to perform subgroup analysis.

For 2-year OS, $BED_{10}$ ($P < 0.001$) was a significant factor for heterogeneity, especially in the low BED group ($P = 0.002$). For the 1-year DCR rate, the number of patients ($P = 0.001$) was a significant factor contributing to heterogeneity, particularly in the larger sample group ($P = 0.004$). None of the parameters affects the heterogeneity for the incidence of overall toxicities.

### Meta-regression analysis

The $BED_{10}$ for 2-year OS and the number of patients for the 1-year DCR rate were included in the meta-regression analysis. The $BED_{10}$ was not identified by meta-regression as contributing to improved 2-year OS ($P = 0.18$, 95% CI [$-0.01$ to $0.01$], $I^2 = 99.79\%$, Adjusted $R^2 = 10.16\%$). However, the number of patients was not associated with a 1-year DCR rate ($P = 0.159$, 95% CI [$-0.02$–$0.01$], $I^2 = 68.0\%$, adjusted $R^2 = 3.03\%$).

## DISCUSSION

The optimal way to treat ICC is through surgery because it results in a more prolonged survival than other treatment modalities. However, many patients have lost the opportunity to accept operations due to the CC's latent beginnings and unusual symptoms. For unresectable CCs, systematic chemotherapy combined with or without local therapy is recommended (*Ohaegbulam et al., 2023*). CC was reported to be radiosensitive. Recently, CC has been reported to be treated using intensity-modulated radiation and three-dimensional conformal radiotherapy (*Sahai & Kumar, 2017*; *Sugiyama et al., 2018*; *Zheng et al., 2018*). The effectiveness of radiation and dosage protection for healthy surrounding tissues still needs improvement due to technical limitations.

As a novel form of radiation, SBRT, based on imaging guidance, utilizes synchronous respiratory tracking technology to track tumors and prevent errors caused by respiratory movement. To treat CC and safeguard normal tissue, SBRT can deliver a suitable therapeutic dose to tumors while minimizing radiation exposure to surrounding healthy tissue. SBRT has already demonstrated significant tumor control when used successfully to treat pancreatic and hepatobiliary malignancies (*Huertas et al., 2015*; *Liu et al., 2021*; *Shampain et al., 2021*; *Zaorsky et al., 2019*). However, there are few reports on unresectable CC. We expected that SBRT might improve the outcome in unresectable CC patients compared to conventional RT with a moderate dose. In a comparative analysis by *Sebastian et al. (2019)* SBRT significantly improved OS in unresectable CC patients relative to conventional locoregional therapies, including transarterial chemoembolization (TACE) and radiofrequency ablation (RFA). The study included 170 patients from the National Cancer Database. However, no other study was found on the pros and cons of SBRT and other treatments. We have to compare the treatment efficacy of SBRT and conventional RT indirectly.

In an older article, patients with unresectable CC who had conventional RT and chemotherapy had 1-year OS rates of 44% and DCR rates of 41%, respectively (*Crane et al., 2002*). Two other retrospective studies of unresectable CC reported 1-year OS rates of 36% (*Zeng et al., 2006*) and 39% (*Chen et al., 2010*), respectively. The pooled 1-year OS and DCR rates in the current meta-analysis were 58.7% and 84.7%, respectively, which were significantly higher than those reported in the above studies. Even the 2-year OS and DCR rates were 33.2% and 70.5%, respectively, which were similar to those reported in the above studies. Therefore, we may conclude that SBRT is a practical treatment approach that works well for individuals with incurable CC, and the toxicity is acceptable. Another vital advantage of SBRT compared with conventional RT is the short delivery time (less than 2 weeks *vs.* 5–6 weeks), which significantly minimizes prolonged breaks from systemic therapy and improves patient compliance (*Zaorsky et al., 2019*).

A recent dose-escalation retrospective research using several RT dosage schemes (median BED, 80.5 Gy; range, 35–100 Gy) revealed that patients with unresectable CC had a median survival of 30 months (*Tao et al., 2016*). These findings support the idea that high-dose RT may have a therapeutic effect for unresectable CC since higher BED dosages were associated with better results. Therefore, $BED_{10}$ was calculated in all the included studies to find the association between $BED_{10}$ and prognosis outcomes. However, our study did not identify a positive association between high BED10 and better clinical outcomes. The reasons could be that most studies were designed retrospectively, and the follow-up times varied. Prospective studies with a large sample size are needed to determine whether CC patients can benefit from a dose-escalation scheme.

Among the included studies, SBRT combined with chemotherapy was the dominant therapeutic approach. The effects of chemotherapy have been documented in patients with CC and those undergoing conventional RT (*Loveday et al., 2018*; *Sumiyoshi et al., 2018*; *Torgeson et al., 2017*). The effects of SBRT combined with chemotherapy were also explored in our study. However, due to the varying chemotherapy regimens and timing of chemotherapy, it is challenging to accurately evaluate the actual value of chemotherapy. In the current investigation, the effects of chemotherapy were evaluated based on whether chemotherapy was administered or not. The effect was not observed in our study. Similarly, prospective studies are needed to determine the preferred chemotherapy regimen, timing, and dose. The role of immunotherapy and targeted therapy should also be evaluated in prospective studies (*Elias et al., 2022*; *Sae-Fung, Mutirangura & Jitkaew, 2022*).

Radiation-associated complications limit the delivery of high-dose RT for CC (*Ben-Josef et al., 2005*). Advanced techniques have been applied to minimize the prescription dose to normal organs, such as image-guided, gated, and tracked modalities. However, these techniques were not used in the included studies. Despite this, the pooled acute, late, and overall toxicity rates were 6.4%, 16.4%, and 16.9%, respectively. A lack of respiratory motion management, in our experience, will significantly increase complications, particularly in hilar tumors. If this motion is not considered, the hilar CC is likely to move significantly during breathing, bringing normal tissue into the high-dose region (*Ben-Josef et al., 2005*; *Sterzing et al., 2014*). Advanced techniques may reduce the risk of complications through precise guidance (*Guckenberger et al., 2008*). Patients with unresectable CC used to have

an extremely poor life expectancy. They would frequently pass away before late toxicity would manifest. Recently, we have noticed an increase in late toxicities in patients with unresectable CC following SBRT. We need to focus more on balancing the rate of late toxicities with irradiated dose levels.

The results of the current meta-analysis should be interpreted with some caveats in mind. First, most of the included studies had a retrospective design. Furthermore, most of these studies were single-center studies with small sample sizes. There could be a significant statistical bias. Second, local lymph node metastatic cases were included in several studies. It is clear from these worse prognoses and shorter survival periods that the accuracy of the overall survival outcomes may be jeopardized. Third, although no publication bias was identified, we noted that the $P$ values for the 1-year and 2-year DCRs were 0.052 and 0.054, respectively. Therefore, publication bias should not be ignored, though it is not significant. Fourth, the chemotherapy scheme was inconsistent among the included studies, which may significantly influence the prognosis. Fifth, some data were extracted from the K-M curves that were not so accurate. Sixth, some vital information, such as OS and toxicity, was unavailable in some studies. Seventh, the influence of tumor location and chemotherapy history on treatment efficacy should be analyzed. However, most studies did not clearly distinguish between them. As a result, it was impossible to analyze the effect of these factors on the efficacy of SBRT.

## CONCLUSION

After OS, DCR, and toxicity data were pooled and analyzed. Our meta-analysis demonstrated that SBRT may be an efficacious and safe therapeutic option for unresectable CC. However, the results should be interpreted cautiously due to high heterogeneity, the retrospective nature of included studies, and the variable chemotherapy used. To determine SBRT's actual function in treating CC, more prospective randomized controlled trials contrasting SBRT with alternative therapy approaches are required.

## ACKNOWLEDGEMENTS

We thank Medjaden Inc. for scientific editing of this manuscript.

### Abbreviations

| | |
|---|---|
| **SBRT** | Stereotactic body radiotherapy |
| **CC** | cholangiocarcinoma |
| **OS** | overall survival |
| **DCR** | disease control rate |
| **RT** | radiation |
| **GI** | gastrointestinal |
| **PROSPERO** | International Prospective Register of Systematic Review |
| **PICOS** | Population, Intervention, Control, Outcomes, and Study Design |
| **BED** | biological effect dose |
| **NOS** | Newcastle Ottawa Scale |

| K-M | Kaplan–Meier |
| ICC | intrahepatic cholangiocarcinoma |

### Funding

The authors received no funding for this work.

### Competing Interests

The authors declare there are no competing interests.

### Author Contributions

- Peng Liu performed the experiments, prepared figures and/or tables, authored or reviewed drafts of the article, and approved the final draft.
- Haiyan Ye performed the experiments, prepared figures and/or tables, authored or reviewed drafts of the article, and approved the final draft.
- Lijun Song analyzed the data, authored or reviewed drafts of the article, and approved the final draft.
- Hua Li analyzed the data, authored or reviewed drafts of the article, and approved the final draft.
- Mingna Fu conceived and designed the experiments, performed the experiments, analyzed the data, authored or reviewed drafts of the article, and approved the final draft.
- Zhichao Dong conceived and designed the experiments, performed the experiments, analyzed the data, authored or reviewed drafts of the article, and approved the final draft.

### Data Availability

The raw data is available in Tables 1 and 2.

### Supplemental Information

Supplemental information for this article can be found online at http://dx.doi.org/10.7717/peerj.19909#supplemental-information.

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
