# Peer review of "Outcomes of stereotactic body radiotherapy for unresectable cholangiocarcinoma: a meta-analysis and systematic review"

_PeerJ, doi:10.7717/peerj.19909_

## Round 0.1 · original submission · Major Revisions

Reviewer 1 ·

Basic reporting

Radiotherapy to unresectable cholangiocarcinoma is a topic that was discussed since many years. The idea is not new. For example, there is a similar systematic review is available (PMID: 30206644) and showed similar results.
This meta-analysis addresses a clinically important question regarding the role of SBRT in unresectable CC. The authors provide a comprehensive and methodologically sound synthesis of available data, concluding that SBRT is a potentially effective and safe treatment option. It is well-conducted with transparent methods. There is comprehensive statistical handling of heterogeneity and clear presentation of results.
The manuscript is well-structured and written in clear, professional English. There are minor typographical errors (e.g., "caDCRulated" instead of "calculated") should be corrected.

Experimental design

The authors should better justify the selection of the BED10 calculation formula and the assumption of α/β = 10, considering variations in tumor biology.
There is inconsistent reporting of heterogeneity for 1-year OS: it is stated that 1-year OS showed significant heterogeneity (I² = 68.6%, P < 0.001). Later, the same outcome is reported as not showing significant heterogeneity (I² = 12.8%, P = 0.316). This discrepancy probably results from different analyses perhaps the second value reflects a sensitivity or subgroup analysis. However, this is not explained clearly in the text.
The conclusions are well-supported by the data but must be interpreted cautiously due to heterogeneity, the retrospective nature of included studies, and variable chemotherapy use.
Missing or inconsistent reporting of chemotherapy regimens may impact the pooled outcomes, especially since chemotherapy likely influences OS and DCR.
Limitations (which are acknowledged by authors): predominantly retrospective studies, variability in chemotherapy regimens, incomplete data in some studies, lack of subgroup analysis by tumor location (intrahepatic vs. extrahepatic).

Validity of the findings

Recommendations:
Clarify the context of heterogeneity findings.
Discuss limitations of BED10 standardization in more depth.
Suggest more explicitly the need for prospective, randomized controlled trials comparing SBRT to conventional radiotherapy or other local treatments.

Additional comments

No comment

Reviewer 2 ·

Basic reporting

The manuscript titled “Outcomes of stereotactic body radiotherapy for unresectable cholangiocarcinoma: a meta-analysis and systematic review” was a meta-analysis study to evaluating the efficacy and toxicities of stereotactic body radiotherapy(SBRT) on unresectable cholangiocarcinoma. In this study, the authors conducted a though comprehensive literature search of Cochrane library, Embase, PubMed, and MEDLINE and finally enclosed 13 studies. After information extraction and synthesis, they found that SBRT for unresectable cholangiocarcinoma obtained an OS was 13.4 months and 1-, and 2-year OS rates were 58.7% and 33.2% respectively, which were no worse than that of chemotherapy

Experimental design

The following issues still need to be mentioned:
1、 AS in LINE112-122 Statistical analysis “…Publication biases were assessed quantitatively using Egger's test of the intercept and visually using the funnel plot…” But, the corresponding content and funnel plot were not found in the results part. The relevant content of Publication bias analysis and funnel plot are strongly recommended to add into the manuscript.
2、 Why there is no pooled result of OS and corresponding forest plot?
3、 The resolution of Fig1,Fig2 is too poor, making it difficult to obtain important information. I suggest that the authors provide more high-quality pictures. In addition, the type of model adopted should be indicated on the forest plot. The title of Figure 3 is incorrect and needs to be modified
4、 Line180-205 Heterogeneity analysis This part is too lengthy and the subtitle does not match the content. It is suggested to simplify and refine it. If necessary, sensitive analysis can be shown with Figures.

Validity of the findings

no comment

Additional comments

Other minor issues need to be mentioned:
1. Line55-56 “However, the treatment effect of conventional RT on CC is suboptimal because t is seldom possible to deliver the ablative dose for tumor control” this sentence is confusing
2. Line58-60 However, a substantial relationship between RT dosage and overall survival (OS) has been proved (Deodato et al., 2006; Morganti et al., 2000). Therefore, more precise RT options are warranted to deliver a high RT dose. Can you explain it more clerely?
3. Line 67-68 However, the treatment effect of SBRT has been controversial in previous systematic reviews (Frakulli et al., 2019; Lee et al., 2019). What controversial effect? Can you explain?
4. Line 68-69 Recently published studies have added to the evidence in this field (Brunner et al., 2019; Kozak et al.,70 2020; Liu et al., 2017; Thuehøj et al., 2022; Zhang et al., 2022). You should state the specific content and data
5. “median overall survival” , “ The median 1-year OS” , “ The median 1-year OS” Appearing many times in the manuscript, Are you imply “overall survival”, “The 1-year OS”, “The 2-year OS” ?
6. Line 150 “The BED10 was caDCRulated for each study.” What does this mean?
7. Line 233 “…CC, the OS is significantly increased compared to transhepatic arterial chemotherapy….” What does the word “”significantly” stand for here? Is this word appropriate used here?

---

## Round 0.2 · Minor Revisions

**Language Note:** When you prepare your next revision, please either (i) have a colleague who is proficient in English and familiar with the subject matter review your manuscript, or (ii) contact a professional editing service to review your manuscript. PeerJ can provide language editing services - you can contact us at [email protected] for pricing (be sure to provide your manuscript number and title). – PeerJ Staff

Reviewer 2 ·

Basic reporting

The revised manuscript titled “Outcomes of stereotactic body radiotherapy for unresectable cholangiocarcinoma: a meta-analysis and systematic review” was a meta-analysis study to evaluate the efficacy and toxicities of stereotactic body radiotherapy (SBRT) on unresectable cholangiocarcinoma. In this study, the authors conducted a thorough, comprehensive literature search of Cochrane Library, Embase, PubMed, and MEDLINE and finally included 13 studies. After information extraction and synthesis, they found that SBRT for unresectable cholangiocarcinoma obtained an OS was 13.4 months, and 1 and 2-year OS rates were 58.7% and 33.2% respectively, which were no worse than that of chemotherapy.

Experimental design

-

Validity of the findings

-

Additional comments

The following issues still need to be mentioned:

1. AS in LINES 142-143, the results of publication bias analysis are explained using text, I still recommend showing the funnel plots as Figures in the manuscript.

2. The resolution of Figures 2 and 3 still needs to be improved, as they are too poor, making it difficult to obtain important information.

---

## Round 0.3 · accepted · Accept

We are pleased to inform you that your manuscript has been accepted for publication. We look forward to receiving your next manuscript.

With best regards,
Yoshi
Prof. Yoshinori Marunaka, M.D., Ph.D.

Reviewer 2 ·

Basic reporting

The revised manuscript titled “Outcomes of stereotactic body radiotherapy for unresectable cholangiocarcinoma: a meta-analysis and systematic review” was a meta-analysis study evaluating the efficacy and toxicities of stereotactic body radiotherapy(SBRT) on unresectable cholangiocarcinoma. In this study, the authors conducted a thorough, comprehensive literature search of Cochrane Library, Embase, PubMed, and MEDLINE and finally included 13 studies. After information extraction and synthesis, they found that SBRT for unresectable cholangiocarcinoma obtained an OS was 13.4 months, and 1-and 2-year OS rates were 58.7% and 33.2% respectively, which were no worse than that of chemotherapy.

The items I have mentioned before have been completely revised and/or explained. I am extremely grateful to the author of the manuscript for their patient responses and meticulous revisions. I believe this manuscript has met the publication requirements of the journal and is acceptable for publication.

Experimental design

-

Validity of the findings

-